# A Review of Talin- and Integrin-Dependent Molecular Mechanisms in Cancer Invasion and Metastasis

**DOI:** 10.3390/ijms26051798

**Published:** 2025-02-20

**Authors:** Zbigniew Baster, Lindsay Russell, Zenon Rajfur

**Affiliations:** 1Institute of Physics, Faculty of Physics, Astronomy and Applied Computer Science, Jagiellonian University, 30-348 Kraków, Poland; 2Laboratory for Cell and Tissue Engineering, Department of Biomedical Engineering, Eindhoven University of Technology, 5600 MB Eindhoven, The Netherlands; 3Undergraduate Program, Barnard College of Columbia University, New York, NY 10027, USA; ler2181@alum.barnard.edu; 4Jagiellonian Center of Biomedical Imaging, Jagiellonian University, 30-348 Kraków, Poland

**Keywords:** cancer, cell migration, talin, integrin, migrastatics

## Abstract

Cancer is the second most common cause of death in the world, representing one of the main economic burdens in health care and research. The effort of research has mainly focused on limiting the growth of a localized tumor, but most recently, there has been more attention focused on restricting the spreading of the cancer via invasion and metastasis. The signaling pathways behind these two processes share many molecules with physiological pathways regulating cell adhesion and migration, and, moreover, adhesion and migration processes themselves underlie tumor potential for invasion. In this work, we reviewed the latest literature about cancer development and invasion and their regulation by cell migration- and adhesion-related proteins, with a specific focus on talins and integrins. We also summarized the most recent developments and approaches to anti-cancer therapies, concentrating on cell migration-related therapies.

## 1. Introduction

In the past years, cancer has been the second most common cause of death in the world after cardiovascular diseases [1,2], and the leading cause in highly developed countries, such as the USA and Western European countries [3]. The development of a cancerous tumor is a multi-step process resulting from the accumulation of multiple mutations and epigenetic alternations, causing the deregulation of cellular functions like proliferation, differentiation, and invasion of the surrounding tissues (Figure 1A) [4,5,6,7]. The cellular microenvironment also plays a crucial role in the onset and progression of cancer. Even temporal non-physiological changes of the microenvironment, caused by events such as trauma or inflammation, may disrupt cellular pathways, promoting cancer progression or even initiating the early stages of carcinogenesis [8]. Invasion is the first step of metastasis and spreading cancer cells into surrounding tissues and lymph nodes [9,10]. During the invasion, cancer cells wade through the extracellular matrix (ECM, see Appendix A. The Extracellular Matrix). To do so, they form actin-rich thin, centrally localized long protrusions on their ventral side called *invadopodia* (Figure 1B) [11,12]. Their primary function is the degradation of the ECM employing various proteases, mainly from the matrix metallopeptidases (MMPs, see Appendix B. Matrix metallopeptidases) family [12,13,14,15], which further allows cells to penetrate the surrounding tissue (Figure 1B) [11].

For this review, we concentrated on selected molecules regulating cell adhesion and migration, focusing on the latest findings describing molecular mechanisms of talin- and integrin-dependent invasion and cancer development. In the first part, we described in detail the structures and main differences between talin isoforms, particularly in their interaction with integrins. We also introduced the molecular mechanisms in which talin–integrin interaction mediates various cellular processes such as adhesion, invadopodia formation, and ECM degradation. In the last part, we described the latest trends in anti-cancer drug therapy development, focusing on the treatments targeting migration- and adhesion-related proteins.

## 2. Adhesion-Related Proteins in Cell Motility

Cell migration and motility underlie many biological processes. These processes include physiological processes, such as wound healing, immunological response, and embryonic and tissue development, and pathophysiological processes, including, mentioned earlier, invasion and metastasis in cancer development [9,21,22,23,24]. There are several different modes (strategies) of cell migration that are regulated by several factors, including cell adhesion level and environmental confinement/crowding [25]. In the case of cancer invasion, there are three most commonly featured modes: two types of single-cell migration, mesenchymal and amoeboid, and collective cell migration [26]. Among them, mesenchymal migration is the most broadly studied mode thus far, especially in research conducted in a high-adhesion environment (Figure 2) [27]. There are numerous proteins involved in the coordination of cell migration [28], including scaffolding [29,30], cytoskeletal [31] and regulatory proteins [32,33], proteases responsible for ECM remodeling [34], or adhesion proteins such as talins and integrins [35,36,37]. In later sections, we concentrate on the latter group of proteins.

### 2.1. Talins

In vertebrates, there are two talin isoforms: talin1 and talin2, which are encoded by the *TLN1* and *TLN2* genes respectively [39]. Thus far, most of the scientific attention has been directed towards talin1 [36].

Talin1 is a large protein [40], described for the first time by Keith Burridge and Laurie Connell in 1983 as a molecule playing a role in focal adhesion dynamics and membrane ruffling [41]. Later, it was shown that talin is crucial for the initiation of cell adhesion by activating integrins [42], and through binding to both integrin and actin, it creates a link between the cytoskeleton and the extracellular matrix [43]. In the following years, multiple binding sites for adhesion- and migration-related proteins were found in talin, including 11 vinculin binding sites [44], a focal adhesion kinase (FAK) binding site [45], and a paxillin binding site [46] (Figure 3A).

Talin1 is composed of two main domains: an N-terminal head FERM (standing for **4**.1, **e**zrin, **r**adixin, and **m**oesin proteins, where it was primarily described [47]) domain [48] and a C-terminal rod domain composed of 13 α-helix bundles [49]. The N- and C-domains are connected by an unstructured linker [37,50]. In general, FERM domains are associated with cytosolic plasma membrane-targeted proteins [47]. Talin1’s FERM domain has an atypical build with an additional F0 subdomain, similar in structure to the F1 subdomain [48]. Thus far, this aberration has been found only in kindlins [48,51]. It is postulated that the F0 domain is specifically required for integrin activation and its stabilization in its active state [52,53], as talin1 and kindlins were shown to be integrin activators [52,54]. In addition to interacting with the plasma membrane and integrins, the head domain has binding sites for several other proteins, including actin (ABS1) [55]. Some of the sites overlap with one another, leading to a complex regulation of talin1’s activity (Figure 3A) [16].

The rod domain also contains multiple binding sites. It has a secondary integrin binding site within R11-R12 bundles [56]; however, the interaction mechanism and its role are still not defined well [36,37]. Moreover, all the talin1’s vinculin binding sites are located in the rod [37,44]. Moreover, the rod has two actin-binding sites [55]. It is postulated that they play different roles in cell adhesion and migration, with one (ABS2) acting as a tension bearer while the other (ABS3) acts as a force-dependent trigger for vinculin binding [36]. Talin1 forms a homodimer through the last C-terminal dimerization helix (DH) [17,37]. Similarly to the head domain, some of the rod’s binding sites overlap (Figure 3A) [16,45,49]. Moreover, mechanical signaling between ECM and cytoskeleton can regulate alternative ligand binding in these sites. For example, upon stretching, the talin1 molecule partially unfolds, exposing vinculin binding sites and, at the same time, disrupting other sites within these regions (Figure 3A) [16].

Talin1’s activity can be regulated through the separation of the head and the rod domains [50,57]. One of the cleavage sites for calpain protease, which mediates talin’s activity, is located in its linker region [36,58]; thus, the site’s conformational availability plays an important role in mediating cell migration and adhesion dynamics [50,58].

As mentioned before, talin1’s functions are associated with the regulation of integrin activity [37,42] and transduction of mechanical cues between the cytoskeleton and the cellular environment [43]. It is also engaged in cell migration [59] and focal adhesion dynamics [50,58,59,60]. Moreover, it mediates invasion [60], invadopodia formation [14], metastasis [60,61], and anoikis [60] in cancer cells.

Talin2 was discovered in 1999 through functional genomic analysis during the search for the third talin1 actin-binding site motif in genomic databases [39,62]. Structurally, talin2 is similar to talin1. Primary structures have shown 76% of identity and 88% of similarity [36,39]. Both proteins also share the same domain and subdomain organization, including the localization of most of the protein interaction sites [36]. The main difference between these two isoforms is in their affinity towards some of the ligands. First, talin2’s head shows a much higher affinity towards integrin than the one of talin1 [63,64]. Interestingly, thus far, it has not been proven that talin2 can activate integrins. In support of this hypothesis, several studies have shown that the depletion of talin2 (opposite to talin1) does not change the integrin activation level during cell spreading [61,63,65]. Furthermore, talin2 also has a higher affinity towards actin [36]. These properties provide molecular context for the observation that talin1 is more abundant in young small peripheral adhesion structures that are highly dynamic, whereas talin2 is rather associated with mature, stable, centrally localized large adhesions [63,66,67]. Although dimerization helices between both talin isoforms are highly conserved, heterodimers have not been described in the literature thus far [36]. Furthermore, the distribution of these two proteins differs among tissues; while talin1 is present in most of the cells in the human body (with the exception of a heart muscle), talin2 is to be found only in selected tissues, such as the skeletal and heart muscles, the brain, and the kidneys [36,68,69,70,71].

The biological role of talin2 is less understood than its sister isoform. As talins have similar structures and share many biological functions [65,72], it was initially presumed that talin2’s functions are redundant with talin1’s [61,65]. However, closer studies have shown that both talin1 and talin2 play distinct roles, and often, both are required in many cellular processes, including tumorigenesis, cancer invasion, and traction force generation [63,73,74]. Furthermore, due to its subcellular central localization, talin2 is believed to have a stronger association with invadopodia maturation and extracellular matrix degradation than talin1 [63]. Moreover, talin2’s muscle tissue specificity and its stronger binding to integrins and actin suggest that one of its distinct roles may be the transduction of forces of a greater magnitude than in the case of talin1 [67,75].

Studies show that loss of talin1 can abrogate cancer invasion [60]. Moreover, several mutations in talin1, which result in the destabilization of its structure, have been associated with cancer development [76]. For example, P229L, R1368W, and L1539P mutants showed decreased recruitment of paxillin and vinculin, showing altered morphology (P229L), migration speed (R1368W—increased, P229L—decreased), invasion rate (R1368W), and proliferation (P229L) [76,77]. Furthermore, L2509P disrupts talin1 dimerization and binding to actin via ABS3, leading to drastic changes in cell and focal adhesions morphology, decreased FAK signaling, paxillin binding, cell migration, and altered proliferation rate [76,77,78,79].

**Figure 3 ijms-26-01798-f003:**
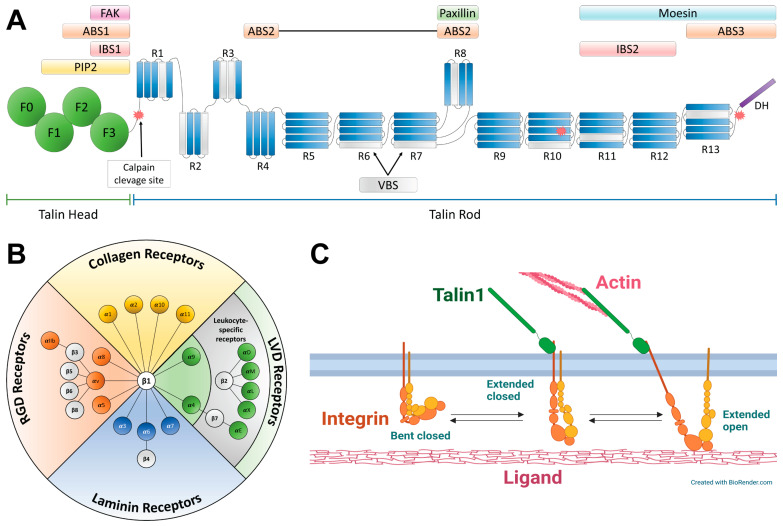
Talins and integrins. (**A**) The structure of talin1 protein. Selected domains, subdomains, and most important interaction sites are marked; vinculin binding regions (VBS) are highlighted in gray. Based on [16,37,44,45,46,48,55]. (**B**) Integrin heterodimers with their most common ligands. Most of them recognize ECM ligands, such as RGD motif (present in e.g., fibronectin) or collagen; a small subgroup of leukocyte-specific receptors recognizes Ig-superfamily cell surface counterreceptors. α4β1 and α9β1 integrins (in green) bind both ECM ligands and the counterreceptors [80]. Based on [80,81]. (**C**) A scheme of the *inside-out* activation of integrin by talin1. Based on [37,54].

### 2.2. Integrins

Integrins are best recognized for anchoring cells to the ECM [80]. They are heterodimeric transmembrane receptors composed of α and β subunits. Most of the subunits have a large extracellular domain responsible for interacting with extracellular ligands, a single transmembrane helix, and a short intracellular domain responsible for interaction with cellular agents [80,82,83]. In mammals, there are 18 α and 8 β subunits, making 24 distinct heterodimer combinations (Figure 3B) [35,80]. Different types of integrins show different affinity towards ECM ligands, depending on their subdomain composition (Figure 3B) [80]. In some cases, the binding site is located on the α subunit; in others, it is shared between both subunits [81].

There are three main integrin conformations connected with the integrin activation stage, and each of them shows a different affinity towards their ligands [37,54]. There are several pathways leading to integrin activation. The most common one, mentioned earlier, is based on an initial interaction with talin1 [37,54]. Unbound integrin resides mainly in a thermodynamically preferable *bent closed* conformation [54,84], which has a very low affinity towards its ligand [84]. Upon the interaction of a β subunit’s cytoplasmic tail with the talin1’s head domain, integrin unfolds, taking an *extended closed* conformation, which still has a low-to-intermediate ligand binding capacity [84]. When interacting with the ligand, integrins are stabilized in the third conformation called *extended open*, which may have an even 5000-fold stronger affinity towards the ligand over the two other states [84,85]. As the signaling comes from the inside of the cell, this kind of activation pathway is called *inside-out* (Figure 3C) [80]. Even though the interplay between talins, integrins, and ECM ligands has been broadly studied, many aspects are still poorly understood, requiring further investigation. One of the recent discoveries showed a nontrivial dependency between a pulling force and an integrin–ligand interaction lifetime called a *catch bond* that stabilizes cell adhesion [35,86].

The second kind of activation pathway, called *outside-in*, is driven by cues from the extracellular environment [80]. Even though the bent closed state is thermodynamically optimal, due to thermal fluctuations, a small amount of unbound integrin is in the extended closed or open conformations (about 0.1% and 0.15%, respectively) [54,87]. Thus, in the extended open conformation, the ECM ligand can be bound simultaneously with talin on the other end, locking integrin in the open state [54]. Furthermore, integrins can be stabilized in the extended conformation by various extracellular (bio)chemical agents like manganese cations or conformation-specific antibodies [84,88].

In addition to playing a key role in cellular adhesion, integrins are involved in many other biological phenomena, including mediation of processes like immune response, cell cycle and proliferation, embryogenesis, and cancer invasion and metastasis [89,90,91,92,93,94]. Furthermore, misregulation of integrin signaling is associated with many pathological processes and diseases [95], such as severe muscular dystrophy (absence of integrin α7) [96], cardiac fibrosis (an overexpression of integrin α11) [97], the leukocyte adhesion deficiency I (a loss of expression of integrin β2) [98], loss of platelet aggregation a (deletion of integrin β3 gene) [96], cancer [99] (e.g., overexpression of αVβ8 promotes growth and invasion of squamous cell carcinoma [100], and overexpression of integrin α11 promotes non-small-cell lung carcinoma [101]).

### 2.3. Molecular Basis of Talin–Integrin Interaction

In the standard model of molecular interactions, called a *slip bond*, the stability of the bond decreases with an exerted pulling force [35]. In the interaction between talins, integrins, and ECM ligands, we can observe the formation of a *catch-slip bond*, or more commonly, a *catch bond* formed between integrins and the ECM. In this kind of intermolecular interaction, the attraction between molecules at first rises together with the force, and then, after reaching a threshold, it weakens [35,72,86]. Thus, binding integrin to the ECM and talin, and further linking the complex to the actomyosin cytoskeleton, provides additional tension that stabilizes integrin–ECM interaction and, therefore, cellular adhesion [54,102]. The mechanism underlying the catch bond in integrins has not been thoroughly described yet [35]. One of the hypotheses presumes that the additional force provided into the interaction stabilizes the fluctuations between the open and closed states of the extended conformation, resulting in an increasing lifetime of the open state (Figure 3C) [35,103]. A follow-up study has supported said hypothesis, showing that upon tension, integrin α5β1 undergoes further conformation changes, leading to the formation of new hydrogen bonds at the interface between integrin and the ECM, thus stabilizing integrin in the open conformation [103,104]. The catch bond behavior was observed for many integrins, including, as mentioned earlier, α5β1 and αVβ3 [35,72,86], the two widely studied RDG-binding integrins.

It is important to underline differences in the molecular mechanisms of interactions between talin1 and talin2 with integrins (in this case, we concentrate specifically on integrin β1). Talins bind to integrin β subunits through its head F3 domain (Figure 3A,C) [37]. The differences in the molecular architecture of the binding sites in talins result in affinity differences between different talin isoforms and integrins, as well as in differences in talin1- and talin2-integrin quaternary structures [63,64,105]. Recent studies have shown that a mutation of just a single residue (C336 or S339 in talin1 or talin2, respectively) is responsible for the majority of the differences [63,64]. Talin1^C336S^ has a higher affinity towards integrin β1 than the wild-type protein, and it has an integrin binding geometry close to talin2^WT^. At the same time, talin2^S339C^ has a lower affinity towards integrins than talin2^WT^ [63,64]. Furthermore, studies made on talin2-knockout cells have shown that the S339C mutant does not rescue the phenotype [63,73]. Therefore, it seems safe to hypothesize that the mutations mentioned above result in differences in the nature of the interaction between talin isoforms and β integrins [64].

### 2.4. Talins and Integrins in the Epithelial–Mesenchymal Transition

Epithelial–mesenchymal transition (EMT) is a process in which epithelial cells lose their epithelial characteristics, like the basal–apical polarity and strong cell–cell adhesion, and acquire mesenchymal–migratory features [106]. As mentioned earlier, high talins and integrins activity results in a decrease in cell-cell adhesion, being one of the driving factors of EMT [107]. Studies have shown that both protein families are the key factors in this process [108].

The talin–integrin complex can activate key EMT pathways, activating focal adhesion kinase (FAK), Src, and the PI3K/AKT and MAPK/ERK cascades [109]. In another EMT-driving mechanism, CdGAP was shown to bind talin and activate integrins in a TGFβ-dependent manner, promoting cell adhesion and TGFβ-induced EMT [110,111].

During EMT, cancer cells can downregulate epithelial-associated integrins, like basement membrane binding integrin α6β4 [112], and overexpress migration-related integrins, like αVβ3 and α1β5 that bind fibronectin, abundant in the interstitial matrix, facilitating the transition to a migratory phenotype [111,113,114,115]. Furthermore, changes in integrin composition stimulate MMPs secretion, promoting ECM degradation and EMT [111,116]. A recent review provided a thorough overview of the role of integrins in epithelial–mesenchymal transition [117].

Interestingly, it also has been demonstrated that a pivotal talin-related mechanism promoting EMT is integrin-independent. Instead, it relies on the interaction between talin and PIPKIγ, promoting mesenchymal traits and inhibiting the expression of E-cadherin, a cell–cell adhesion-related protein, in cancer cells [108,118].

### 2.5. Interplay Between Talins and β1–Integrin in Invadopodia Formation and Maturation

As mentioned earlier, *invadopodia* are actin-rich protrusions directed towards the extracellular matrix. Their main task is to degrade and penetrate the neighboring matrix to allow cell invasion and metastasis (Figure 1B, insert) [13]. The formation of an invadopodium starts with the assembly of precursors, such as cortactin, cofilin, Arp2/3, and N-WASp, that are later anchored to the plasma membrane by the Tks5 protein [119,120]. In the next step, β1 integrin is recruited to the complex [13,121]. In the late maturation stages, invadopodium continues to elongate based on actin polymerization [13]. In these stages, microtubule filaments are also found in these protrusions, presumably serving as trafficking routes for proteases-containing vesicles (such as MMP2, MMP9, or MT1-MMP, see Appendix B. Matrix metallopeptidases) [34,122].

In contrast to focal adhesion formation, talin1 binds to the invadopodium precursor complex independently from β1 integrin [13,14]. Nonetheless, further interaction between these two proteins is critical for the recruitment of the moesin–NHE-1 complex, which leads to the initiation of degradation of the ECM by stimulating membrane type 1 matrix metalloproteinase (MT1-MMP) [13,14]. Simultaneously, cofilin activation promotes actin polymerization and growth of the invadopodium [13,14,121,123]. Interestingly, recent studies have shown that talin2, through its interaction with integrins, also mediates the maturation of invadopodia, yet it is involved in a distinct pathway [13,63,74].

Both talins were shown to co-localize with Tks5 at the invadopodium-precursor site, which suggests their involvement in invadopodia formation. Beaty and colleagues [14] showed that talin1 binding to invadopodia is independent of β1 integrin, but it is mediated by actin binding via ABS3 (see Figure 3A). Nonetheless, talin1 interaction with β1 integrin is crucial for further invadopodia maturation. Interestingly, this process is mediated not by the main integrin binding site in the talin1 head domain but by the secondary IBS2 site in the R11 rod section of the protein, as a re-expression of talin1 rod domain rescued talin1 depletion, but integrin binding deficient mutant of talin1 rod, as well as talin1 head domain did not [14].

On the other hand, MDA-MB-231 breast cancer cells and U-2 OS osteosarcoma talin2-depleted cells show inhibition in ECM degradation and invadopodia formation, even in the presence of talin1 [63,74]. Intriguingly, the re-expression of talin2^S339C^ mutant, having an altered nature of talin2–integrin interaction (see Section 2.2), does not rescue this process [63,73,74], suggesting specific talin2–integrin interaction. Further, it was shown that depletion of talin2 inhibits the secretion of MMP9 by reducing docking of MMP9-containing vesicles to the cell ventral membrane, yet the complete mechanism of the process is still to be uncovered [74].

Importantly, depletion of either talin resulted in inhibition of tumor growth and invadopodia formation, but depletion of talin2 seems to have a more significant effect [63,73]. This implies that talin1 and talin2 play separate, non-redundant roles in cancer development.

### 2.6. Talins and Integrins in Cancer Cells–Tumor Microenvironment Interaction

Tumor microenvironment (TME) is one of the most critical factors in the regulation of cancer invasion and metastasis [8,27]. Biochemical, cellular, structural, and mechanical signaling from the microenvironment influences cancer cell migration, adhesion, invasion, proliferation, angiogenesis potential, and many other cellular properties driving carcinogenesis [27,107,124]. Moreover, the tumor microenvironment can promote or suppress carcinogenic features, leading to high heterogeneity, both within a single tumor and between tumor sites [8]. The TME evolves and changes together with the development of the tumor itself [107]. Its remodeling, in a significant part, is driven by cancer cells that can deposit, proteolytically degrade, and post-translationally modify ECM proteins, as well as physically remodel the ECM organization [125]. Talins and integrins are directly involved in the synthesis of the ECM, regulating secretion and reorganization at a molecular level of the ECM’s components, such as collagens and fibronectin [126,127]. As mentioned before, they also regulate protease secretion, leading to degradation of the ECM [74]. Both processes lead to physical reorganization of the TME, usually leading to changes like the stiffening of the ECM or remodeling of collagen into straight bundles [128,129]. All these changes lead to further deviation of TME from the physiological state and create a self-propelling mechanism in which more integrins become engaged, further remodeling the ECM and simultaneously activating FAK/Src signaling and promoting cell survival mechanisms. Additionally, this mechanism reinforces cell-ECM adhesion, leading to increased proliferation and overcoming cell–cell adhesion, causing detachment of single cells, therefore promoting invasion [107,129,130,131]. Moreover, cancer cells, via integrin-based adhesions, can physically reorganize and align collagen fibers by generating contractile forces [132]. This way, cells can form the pathologically straight ECM fibers architecture that facilitates cell polarization, directed cell migration, and metastasis [128].

Multiple studies have shown the importance of integrins in interacting with TME. As mentioned earlier, dysregulation of integrins expression leads to a change in tumor cells’ preferential ligand supporting EMT [111,115]. The interaction between β1 integrin and talin 2 promotes the secretion of MMPs in breast cancer, leading to ECM remodeling and degradation [74,133]. Interestingly, tissue inhibitor of metalloproteinases 2 (TIMP2, see Appendix B. Matrix metallopeptidases) can bind directly to integrin α1β3, inhibiting angiogenesis [134]. Overexpression of talins, integrins, and other adhesion-related proteins can mimic some part of the integrin-based ECM adhesion-signaling in circulating tumor cells (Figure 1A *Dissemination*), promoting FAK activation and its downstream effectors, inducing cell survival and resistance to anoikis (programmed cell death resulting from detachment from the ECM) [60,135,136,137]. Moreover, studies have shown that talin1-mediated anoikis resistance can be independent of its interaction with integrin [60,136].

Interestingly, knocking down talin 1 or treatment with cyanidin-3-glucoside, a talin–integrin- interaction-targeting natural compound, inhibited the growth of HT-29 cancer micro-tumors [137], though the mechanism of this process is not well described [138]. Furthermore, the talin–integrin complex promotes activation of the FAK/Sac pathway, promoting tumor growth in situ [116,139,140].

## 3. Clinical Aspects of Talin and Integrin in Cancer Development

### 3.1. Talin- and Integrin-Based Cancer Prognosis

Multiple studies have shown a correlation between talins’ and integrins’ expression levels in tumors and both cancer development and patients’ survival [138,141]. Our recent study showed that talin2 is upregulated in many cancer types, including pancreatic adenocarcinoma (PAAD), cholangiocarcinoma (CHOL), stomach adenocarcinoma (STAD), lung squamous cell carcinoma (LUSC), prostate cancer (PRAD), and liver hepatocellular carcinoma (LIHC) [142]. For the current work, we reviewed several studies [143,144,145,146,147] included in the Kaplan–Meier Plotter database [145] and analyzed the dependence of hazard ratio (HR) on high expression levels of talins and integrins (except for integrin α1, not included in the database) (Figure 4A). Interestingly, different cancers showed different prognoses based on the high expression of these proteins. This supports seemingly contradictory studies that have shown that potential anti-cancer drugs mediating talin–integrin interaction can have adverse effects on cellular processes, such as adhesion, in different cancer types as showed in [138,148], further underlining the complexity in the regulation of these two protein families’ activities. Increased levels of talin1, integrins α5-8, α10, α11, αV, β3-5, and leukocyte-specific integrins α4 and αE show correlation with either high increased or high decreased HR in various cancers, making them potential candidates for treatment prognosis markers and therapy targets. Interestingly, most cancers of lower survival rates (acute myeloid leukemia (AML), ovarian, and lung cancers) show a lower correlation of the level of studied proteins to HR (Figure 4A), suggesting long-timescale effects of talin- and integrin-regulation during cancer progression.

### 3.2. Integrin-Related Immune Evasion and Anti-Cancer Drug Resistance

Integrins, as surface receptors, regulate cell–immune cell interaction. Moreover, some integrin heterodimers are specific to immune cells (Figure 3B). In liver and colon cancers, elevated expression of ICAM-1, VCAM-1, and MAdCAM-1, ligands to leukocyte-specific integrins, have shown improved T-cells penetration of the tumor and prognosis on patients’ survival [152,153]. On the contrary, other studies have shown that the upregulation of VCAM-1 and its interaction with integrin α4β1 promoted angiogenesis, invasion, and tumor progression in neuroblastoma and gastric cancer [154,155]. Another integrin, αVβ6, is essential in tumor development [156,157]. Recent studies have shown that overexpression of integrin αVβ6 inhibits T-cell anti-tumor response via TGF-β–SOX4 pathway, providing an efficient immune evasion strategy for cancer cells. Treatment with a blocking anti-integrin αVβ6 antibody inhibited tumor progression and promoted T-cell immunoresponse, showing a promising new target to enhance current therapies [158,159]. Similar observations were made for integrin αVβ8, also acting through the TGFβ pathway [160], and for integrin αVβ6 through the promotion of PD-L1 expression [161]. Furthermore, integrins regulate response to anti-tumor therapies [162]. It was shown that integrin αVβ3, acting through the KRAS–RalB–NF-κB pathway, stimulated EGFR inhibitor resistance [163]. Moreover, the α6 integrin–Src–Akt pathway and β1 integrin in a GPER-dependent pathway induce tamoxifen resistance [164,165]. Several more specialized reviews have recently presented broader overviews of this topic [149,162,166].

### 3.3. Migrastatics

Although the FDA approves about 15 new cancer treatments every year, which is approximately 30% of all new annually approved treatments, over the years, the number of deaths caused by cancer has constantly been rising (Figure 4B,C, Appendix C. Novel anticancer treatments: Table A1) [150,151,167]. It is important to underline that this trend is significantly impacted by better diagnostic methods and society’s aging, which is one of the primary cancer risk factors. However, it also shows that the current approach to new anti-cancer drug development is not efficient enough.

Most of the current therapies concentrate on reducing cancer development at the tumor site, regardless of whether it is a primary or secondary location. They aim to inhibit cancer cell proliferation and reduce tumor size, which for a long time was a requirement for FDA approval. On the other hand, currently, only a few anti-cancer therapies are aiming at metastasis, even though data show that it is related to over 60% of all cancer-related deaths [168].

In 2018, the FDA approved a new endpoint in clinical trials: metastasis-free survival, which allows evaluation of the effectiveness of an anti-cancer therapy based on the formation of metastatic tumors [169]. Around the same time, in 2017, a new term was coined called *migrastatics* for drugs that aim to hinder the invasiveness of cancer cells and reduce their ability to metastasize [170]. It is important to note that these drugs are meant to complement antiproliferative therapy rather than replace it. As cell migration has been intensively studied within the past decades, many drugs used in vitro may be suitable as migrastatics. The main drawback, however, might be high toxicity to healthy cells [170,171]. Currently, several clinical trials are aiming directly at cancer metastasis [170,171].

### 3.4. Talin and Integrin as Targets for Anti-Cancer Therapies

As mentioned earlier, integrins are an emerging target for anti-tumor therapies. Currently, there are seven integrin-targeting drugs on the market, yet their primary use is in cardiovascular and other non-cancer-related therapies [172]. Moreover, anti-cancer use of integrin-related therapeutics has not been approved so far. By now, there have been at least 230 clinical studies on therapies targeting various integrins, including anti-cancer therapies. These treatments primarily aim to target integrin subtypes associated with the development and progression of various tumor types [173]. For example, Etaracizumab (MEDI-522) is believed to be able to target integrin αvβ3 in selected tumors. It has been studied through several Phase I and II clinical trials [173,174,175]. Moreover, a substantial part of cancer-related integrin clinical research is focused on targeting integrins for techniques such as positron emission tomography (PET) imaging [176].

At the moment, a new wave of studies is recruiting for various forms of integrin-targeted drugs specifically geared towards cancer treatment. As mentioned in the previous paragraph, integrin targeting can be the aim of both molecular imaging techniques supporting chemotherapy as well as a part of the therapy itself. For example, the group of Hao Wang is studying the application of integrin ligand-bound Fluor-18 in PET imaging [177,178], and the Sutcliffe group is aiming to use an integrin αVβ6-targeting drug to deliver therapeutical Lutetium-177 radionuclides to the tumor site [179], respectively. Several recent studies from 2022 have provided detailed overviews of completed and ongoing integrin-targeting clinical trials [172,180,181].

In several cases, currently approved drugs have shown a potential integrin-regulation function. For example, Levothyroxine, a synthetic T4 hormone used in addition to traditional radiation and chemotherapy in the treatment of thyrotropin-dependent well-differentiated thyroid cancer [182], was also shown to activate integrin αVβ3 [183]. However, to our knowledge, its integrin-activation-related properties have not yet been addressed in any clinical study.

Currently, there is little focus on talin-targeted cancer treatments at the clinical studies stage. Yet, several cellular-level studies have shown talin or the talin–integrin interaction as a potential future target for clinical studies. For example, docetaxel showed promising results inhibiting talin2 expression in a gastric cancer MKN45 cell line [184]. Moreover, several in vitro studies suggest that anthocyanins, found in natural products like dark fruits and vegetables, are potential regulators of talin–integrin interaction that leads to decreased tumor growth and cancer invasion [138,185].

## 4. Conclusions

Interaction between cells and their environment drives processes in human bodies. Here, we wanted to underline its importance in cancer development, presenting current knowledge on two main protein families responsible for mechanosensing of the cellular environment: talins and integrins, in the context of cancer metastasis and development. Many of the factors that influence the risk of developing tumors also influence cell–ECM interaction, and biochemical and mechanical properties of the ECM [136,138]. Changes in the cell–environment interaction underlie cancer invasion and metastasis, which is one of the hallmarks of cancer [10], poorly projecting on expected patients’ survival [168]. Many of the proteins involved in interaction with ECM are also directly involved in the initiation of cancer invasion [14,74,121].

The recent changes in FDA policy on anti-cancer therapies [169] are one example of how the scientific community is shifting its attention towards targeting mechanisms mediating cancer metastasis. In this work, we introduced and summarized recent studies describing some of the molecular mechanisms of interaction between talins and integrins that may lead to invasion and cancer progression and the latest advances in clinical research targeting cancer metastasis. We believe that further exploration of this and other cell adhesion and migration-related pathways in the context of regulation of carcinogenesis is crucial for developing new and better anti-cancer therapies in the future.

## Figures and Tables

**Figure 1 ijms-26-01798-f001:**
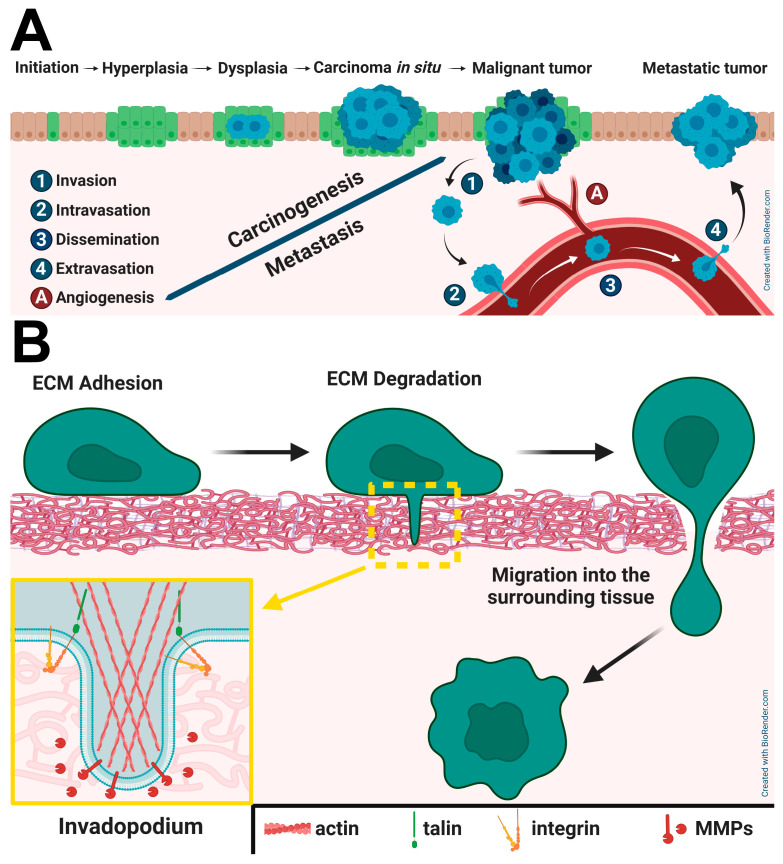
Stages of cancer development. (**A**) Tumorigenesis in an epithelial layer. In the initial stage of carcinogenesis, accumulated mutations cause deregulation of cell growth and differentiation, what leads to uncontrolled cell division and finally to hyperplasia. Further DNA damage causes loss of cells’ classical morphology in a dysplasia stage. Though dysplasia does not ascertain the development of cancer, in some cases altered cells may eventually occupy the entire cellular layer and create carcinoma in situ. Invasion, the detachment of a cell from the primary tumor site, initiates the process of creation of secondary tumors called metastasis. After penetrating the surrounding tissue, a cancer cell can enter the circulatory system through the process of intravasation. Pathological angiogenesis supports this phenomenon further, making blood vessels more accessible. Circulating tumor cells can disseminate at distant sites of the body [16,17]. Then, through the process of extravasation, cells leave blood vessels and find new niches in remote tissues to develop secondary (metastatic) tumors. Based on [6,7,18,19]. (**B**) Stages of cancer cell invasion from epithelial tissue. During this process, cell-cell interactions weaken, and cell-ECM interactions become stronger. In the second stage of invasion, in order to wade through the ECM, cancer cells form invadopodia, allowing them to penetrate to the surrounding tissue in the last step of invasion. Insert: a simplified scheme of an invadopodium. Based on [11,12,13,20].

**Figure 2 ijms-26-01798-f002:**
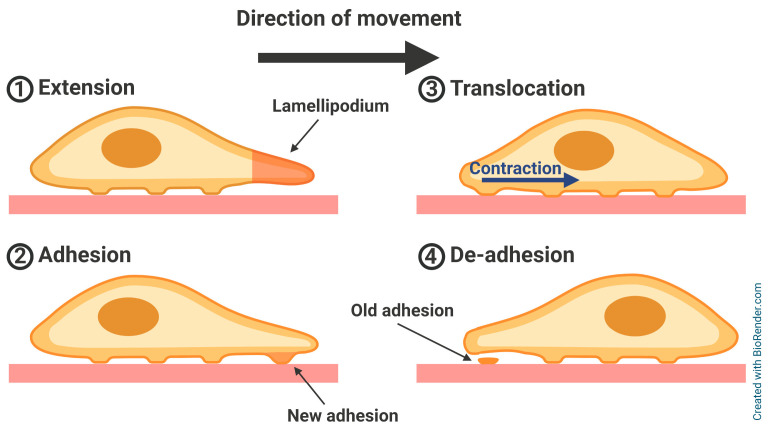
The scheme of a single-cell mesenchymal migration mode. (**1**) In the first step, the cell protrudes a wide projection at the leading edge called the *lamellipodium*. (**2**) At the interface between the lamellipodium and the substrate, new adhesion structures are formed to stabilize the new position. (**3**) The contraction of the actomyosin cytoskeleton creates a force that propels the cell body towards the leading edge. (**4**) Adhesions in the back of the cell disassemble to allow retraction of the cell’s tail. Then, the cell can repeat the cycle. Based on [38].

**Figure 4 ijms-26-01798-f004:**
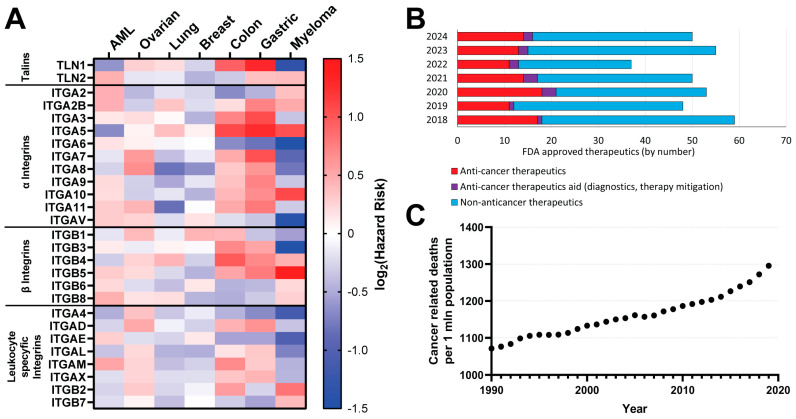
(**A**) The heat map of the average hazard ratio in relation to high expression of talins and integrins in patients suffering from various types of cancers, based on Kaplan–Meier plots. Based on [143,144,145,146,147]. (**B**) Number of FDA-approved therapies in the past years with a focus on anti-cancer therapies. Data from [149]. (**C**) Cancer-related deaths in the past 30 years per 1 mln population. Data based on the Global Burden of Disease Database [150,151].

## Data Availability

The data supporting this study’s findings are publicly available from referred sources. The processed data are available from the corresponding author upon reasonable request.

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
