# Peer review of "A Review of Talin- and Integrin-Dependent Molecular Mechanisms in Cancer Invasion and Metastasis"

_ijms, 2025, doi:10.3390/ijms26051798_

Round 1
Reviewer 1 Report
Comments and Suggestions for Authors
A latest review focus on the development and invasion of cancer, regulated by cell migration- and adhesion-related proteins, particularly emphasizing the roles of talins and integrins. The review also highlights the latest advancements and strategies in anti-cancer therapies, with a special emphasis on therapies targeting cell migration. Several key points should be noted as follows:
1) A diagram illustrating "Molecular Regulation of Talin and Integrin in Cancer" would help clarify this issue.
2) EMT in important for cancer invasion and metastasis. Lots of studies have bee done about “Talins and integrins in the epithelial-mesenchymal transition”. However, the content in this aspect is not very elaborate (line 325-343). More details should be provided.
3) From Fig. 2A, are there any studies on the role of Talin and Integrin in the progression from initiation to CIS?
4) About “cancer”, “cancer cells - tumor microenvironment (TME)interaction”, one paper is suggested to be reviewed. This paper proposes that cancer is not a genetic disease but an ecological disease: a multidimensional spatiotemporal "unity of ecology and evolution" pathological ecosystem (https://pubmed.ncbi.nlm.nih.gov/37056571/). This view might help for our understanding of cancer complex causal process including interacting with TME.
5) As far as we know, Bissell MJ has carried out numerous pioneering research endeavors in this respect (TME). Her research commenced as early as the 1980s and was primarily centered on ECM including Integrin and carcinogenesis. Should the author have failed to notice this research aspect, supplementation is advisable.
Author Response
Comment 1: A diagram illustrating "Molecular Regulation of Talin and Integrin in Cancer" would help clarify this issue.
Response 1: We added a graphical abstract showing the overview of "Molecular Regulation of Talin and Integrin in Cancer".
Comment 2: EMT in important for cancer invasion and metastasis. Lots of studies have bee done about “Talins and integrins in the epithelial-mesenchymal transition”. However, the content in this aspect is not very elaborate (line 325-343). More details should be provided.
Response 3: We expanded and reorganized the paragraph. Additionally, we recommend a review that explores the role of integrins in EMT in-depth. We believe we underlined key EMT pathways that depend on talins and integrins. Furthermore, EMT is closely related to adhesion and interaction with the ECM, which we discussed in other sections of the manuscript.
Comment 3: From Fig. 2A, are there any studies on the role of Talin and Integrin in the progression from initiation to CIS?
Response 4: There are several studies (including one of ours) showing a correlation between talin and integrin activity and CIS. We added a short paragraph to “Talins and integrins in cancer cells - tumor microenvironment interaction section overviewing” this topic.
Comment 4: About “cancer”, “cancer cells - tumor microenvironment (TME)interaction”, one paper is suggested to be reviewed. This paper proposes that cancer is not a genetic disease but an ecological disease: a multidimensional spatiotemporal "unity of ecology and evolution" pathological ecosystem (https://pubmed.ncbi.nlm.nih.gov/37056571/). This view might help for our understanding of cancer complex causal process including interacting with TME.
Response 4: Thank you for introducing this interesting paper. We included it in our manuscript and further underlined the importance of TME in tumor development, both in the introduction and in the “Talins and integrins in cancer cells - tumor microenvironment interaction” section.
Comment 5: As far as we know, Bissell MJ has carried out numerous pioneering research endeavors in this respect (TME). Her research commenced as early as the 1980s and was primarily centered on ECM including Integrin and carcinogenesis. Should the author have failed to notice this research aspect, supplementation is advisable.
Response 5: We expanded “Talins and integrins in cancer cells - tumor microenvironment interaction” and included references to works of Bissell MJ.
Reviewer 2 Report
Comments and Suggestions for Authors
This review synthesizes a large number of recent studies related to tumor cell motility and provides a clear overview of this complex topic. The authors focus on the roles on talins and integrins in invasion and metastasis, and thus provide a fairly unique perspective on these critical stages of tumor evolution. The sections are logically organized and generally well written.
A few minor edits are recommended . The status of cancer as a leading cause of death is widely recognized and has been extensively reviewed. It seems somewhat perfunctory to discuss the epidemiology of cancer in the context of talins and integrins. Omitting these parts from the introduction, along with figures 1 and 5c, would make the article more focused and therefore more useful. Otherwise, the figures are well rendered and highly illustrative.
Author Response
Comment 1: A few minor edits are recommended . The status of cancer as a leading cause of death is widely recognized and has been extensively reviewed. It seems somewhat perfunctory to discuss the epidemiology of cancer in the context of talins and integrins. Omitting these parts from the introduction, along with figures 1 and 5c, would make the article more focused and therefore more useful. Otherwise, the figures are well rendered and highly illustrative.
Response 1: Thank you for the suggestion. Our submitted version of the manuscript developed from its initial concept, and at first, we reduced the epidemiologic content. We agree that the epidemiologic part is superfluous in the current version of the manuscript. We reduced the introduction and removed the Supplementary Note and Figures 1, S1, and S2. We decided to keep Figure 5C (Now 4C after revision) as it presents the issue of a growing cancer death toll that we want to underline.
Reviewer 3 Report
Comments and Suggestions for Authors
This manuscript review the recent progress of talin and Integrin in cancer. This work provides information to the audiences in this field. But the work need to be re-organized before acceptance.
1. For a review paper, I don't think the supplementary materials is necessary. Most of supplementary figures and tables are not tightly related to the main content. So it could be removed.
2. line 175, should be 2.2 integrins, and so on....
3. line 325, 2.5(should be 2.6) Talins and integrin in the EMT should be moved to as 2.4. Because it is a in vivo normal cell progress. and then line 254 2.3 (should be 2.4)
4. line 454, for a review paper, it is very strange to list a discussion session. just conclusion is ok.
5. Appendix A and B should provide 2 figures
Author Response
Comment 1: For a review paper, I don't think the supplementary materials is necessary. Most of supplementary figures and tables are not tightly related to the main content. So it could be removed.
Response 1: Thank you for the suggestion. Our submitted version of the manuscript developed from its initial concept, and at first, we reduced the epidemiologic content; we agree that in the current version of the manuscript, the epidemiologic part is superfluous. We reduced the introduction and removed the Supplementary Note and Figures 1, S1, and S2. We kept the other supplementary figures, as we find them relevant, yet we do not see the right way to include them in the main manuscript.
Comment 2: line 175, should be 2.2 integrins, and so on....
Response 2: Subsection numbering corrected
Comment 3: line 325, 2.5(should be 2.6) Talins and integrin in the EMT should be moved to as 2.4. Because it is a in vivo normal cell progress. and then line 254 2.3 (should be 2.4)
Response 3: Thank you for this suggestion. We agree that this provides a more logical flow to the text, and we changed it as suggested.
Comment 4: line 454, for a review paper, it is very strange to list a discussion session. just conclusion is ok.
Response 4: When we were preparing the manuscript, we tried to fit the journal’s template, which includes a discussion section, even though we also found it not entirely suitable. We are happy to have support in this matter. We corrected the section title.
Comment 5: Appendix A and B should provide 2 figures
Response 5: The appendices we provided aim to introduce important topics that provide context and highlight further complications of the regulation of cancer metastasis in relation to the regulation of cancer progression by talins and integrins. Including them in the manuscript helps place our review within a broader overview; however, they are less directly connected to the main topic than other sections. Therefore, we do not believe that additional figures are necessary, and the brief description we provided is adequate.
Round 2
Reviewer 1 Report
Comments and Suggestions for Authors
The authors have made very good revisions based on my review comments.
Author Response
Thank you for your advice and time.
Reviewer 3 Report
Comments and Suggestions for Authors
This revision has been partly modified based on my suggestions. But there are 2 points that I suggest the authors to further revise the current version:
1. Supplemental figure 1--8 should be removed from this manuscript. I really don't know why the authors keep them in a review paper?!
2. I still suggest the authors provide 2 figures corresponding to Appendix A and B, respectively.
It will be greatly helpful for the audience to understand ECM and MMPs. Text description is really bad choice.
Author Response
Comment 1: Supplemental figure 1--8 should be removed from this manuscript. I really don't know why the authors keep them in a review paper?!
Response 1: We removed Figures S1-8, and moved Table S1 as "Appendix C. Novel anticancer treatments" removing this way all supplementary materials.
Comment 2: I still suggest the authors provide 2 figures corresponding to Appendix A and B, respectively.
Response 2: We included 2 figures to Appendices A and B. Additionally in "Talins and integrins in cancer cells - tumor microenvironment interaction" section, we added additional information about interaction between TIMP2 and integrins that we found during figures preparation.
Round 3
Reviewer 3 Report
Comments and Suggestions for Authors
I have no further questions on the current version